# Do We Need a Specific Corpus and Multiple High-Performance GPUs for Training the BERT Model? An Experiment on COVID-19 Dataset

Nontakan Nuntachit [1,2] and Prompong Sugunnasil [3,*]

1   Data Science Consortium, Faculty of Engineering, Chiang Mai University, Chiang Mai 50200, Thailand; nontakan_nuntachit@cmu.ac.th
2   Department of Internal Medicine, Faculty of Medicine, Chiang Mai University, Chiang Mai 50200, Thailand
3   College of Art, Media, and Technology, Chiang Mai University, Chiang Mai 50200, Thailand
*   Correspondence: prompong.sugunnasil@cmu.ac.th; Tel.: +668-1672-1282

**Abstract:** The COVID-19 pandemic has impacted daily lives around the globe. Since 2019, the amount of literature focusing on COVID-19 has risen exponentially. However, it is almost impossible for humans to read all of the studies and classify them. This article proposes a method of making an unsupervised model called a zero-shot classification model, based on the pre-trained BERT model. We used the CORD-19 dataset in conjunction with the LitCovid database to construct new vocabulary and prepare the test dataset. For NLI downstream task, we used three corpora: SNLI, MultiNLI, and MedNLI. We significantly reduced the training time by 98.2639% to build a task-specific machine learning model, using only one Nvidia Tesla V100. The final model can run faster and use fewer resources than its comparators. It has an accuracy of 27.84%, which is lower than the best-achieved accuracy by 6.73%, but it is comparable. Finally, we identified that the tokenizer and vocabulary more specific to COVID-19 could not outperform the generalized ones. Additionally, it was found that BART architecture affects the classification results.

**Keywords:** COVID-19; CORD-19; LitCovid; BERT; zero-shot learning; classification; unsupervised machine learning; deep-learning; BART; RoBERTa; SciBERT; Natural Language Processing

## 1. Introduction

Since the outbreak of COVID-19, not only has the number of patients increased every day, but the number of studies about this disease has also increased exponentially. It is time-consuming for health-care persons to read all of this literature to find a study that matches their needs. To collect information about COVID-19 literature, the Allen Institute of AI and its partners have released a dataset, CORD-19 [1], that aims to connect the medical and machine learning community to find a solution to this outbreak. Until now, 1669 projects have used this dataset on the Kaggle website to try to find a solution, but there are none that use the "zero-shot learning" method [2,3].

Zero-shot learning is a machine learning method that can predict unseen classes during the testing phase using additional encoding information to distinguish the type of objects. The first study to use this method was introduced in 2008, but it was named "dataless classification [4]". This paper focused mainly on classifying text documents from the newsgroups dataset and Yahoo! Answer. So far, data scientists are able to use this method in Natural Language Processing (NLP) and Computer Vision (CV) [5].

Moreover, there is a new approach for NLP called "attention" [6]; this technique was introduced in 2017 to overcome the limitations of earlier work, such as ELMo [7], which uses a bidirectional Long-Short term Memory (LSTM) model [8]; this model allowed scientists to overcome the inability to take both the left and right contexts of the target word when calculating the meaning of that word. In this work, each word input is embedded into

a vector, and then calculated into a Queries, Keys, and Values matrix. Finally, the attention score is calculated by the softmax value function of the dot product of matrix Q, with the transpose of matrix K divided by the square root of key numbers and multiplied with matrix V, where $\sqrt{d_k}$ is the dimension of the key vector, to obtain the final attention score. This model is called a "Transformer" [6] model and consists of encoder and decoder functions.

$$\text{Attention}(Q, K, V) = \text{softmax}\left(\frac{QK^T}{\sqrt{d_k}}\right) V \tag{1}$$

### 1.1. Related Works

1.1.1. BERT: Pre-Training of Deep Bidirectional Transformers for Language Understanding

In 2019, Google released a new language model, "BERT" [9], which stands for **B**idirectional **E**ncoder **R**epresentations from **T**ransformers. BERT uses the encoder part of a transformer model [6]. The development of BERT consisted of two steps: the semi-supervised and supervised stages. In the first stage, BERT was trained to comprehend a linguistic context using two semi-supervised tasks on a particular corpus. The Masked Language Model (MLM) was the first semi-supervised task. In this task, the model attempts to guess the hidden words by masking 15% of the words in the sentence. The second semi-supervised task was Next Sentence Prediction (NSP). Whether sentence B comes after sentence A can be predicted by the model. The second stage of the training task depends on the task we want to use, ranging from classification to question-and-answer.

Since the technique involves a large number of data and requires a tremendous computational resource, numerous models based on this technique have started to rely on the optimization used to speed up training after Google released the BERT model. The BERT variants based on optimization are listed in Table 1.

**Table 1.** Example of BERT model and its variants based on optimization.

| Model | Full Term | Summary |
|---|---|---|
| BERT-Base BERT-Large | Bidirectional Encoder Representations from Transformers | BERT-Base (L = 12, H = 768, A = 12, Total Parameters = 110M) BERT-Large (L = 24, H = 1024, A = 16, Total Parameters = 340M) L is the number of layers (i.e., Transformer blocks), H is the hidden layer, and A is the number of self-attention heads. |
| ALBERT [10] | A Lite BERT | The ALBERT model has 12 million parameters with 768 hidden layers and 128 embedding layers |
| RoBERTa [11] | Robustly Optimized BERT pre-training Approach | Uses different pretraining step <ul><li>Dynamic Masking</li><li>Removes Next Sequence Prediction (NSP) task</li><li>Has more data points; trained on Common-Crawl News and OpenWebText</li><li>Large batch size: batch size of 8000 with 300,000 steps</li></ul> |
| DistilBERT [12] | Distillation BERT | Uses the knowledge distillation method to reduce the model size and achieve 97% of the ability of the original BERT |

Some BERT-based models aim to be used in the biomedical field. However, the techniques and input used in each model are quite different, as the objective of each model is not the same. Table 2 compares BERT-based biomedical models, including our model, from multiple perspectives.

**Table 2.** Comparison of biomedical BERT-based models.

| Criteria | PubMedBERT [13] | Clinical BERT [14] | Med-BERT [15] | Our Model |
|---|---|---|---|---|
| Type of corpus | Articles from PubMed | Clinical notes | ICD-9 + ICD-10 code for diagnosis | CORD-19 dataset |
| Vocabulary size | Same as original BERT (30 K) | N/A but taken from 2 million clinical notes | 82 K | ~33 K |
| Pretraining data source | Wikipedia, BookCorpus, PubMed articles | MIMIC-III v1.4 [16] | General Electronic Health Record (EHR) | Computer science and broad biomedical papers |
| Data source size | 14 million abstracts, 3.2 billion words (filter from 4 billion words), 21 GB | 26 tables in a relational database, 23.5 GB compressed zip file | 28 million EHRs | 3.4 GB (filter from 38.91 GB raw data) |
| Input data structure | Biomedical vocabulary | Clinical notes, MedNLI | ICD Code + visit + serialization embeddings | COVID-19 article abstract, SNLI, Multi-NLI, MedNLI |
| Pretraining task | From scratch | From scratch, initialized from BioBERT [17] | Masked LM + predict length of stay | Adding new vocabulary and creating an NLI downstream task |
| Evaluation task | NER, PICO, relation extraction, sentence similarity, document classification, question answering | Clinical NLP tasks | Disease predictions | Same as SciBERT [18] + Zero-shot classification |

Most BERT-based models were trained on a specific corpus for use in a different context, and most of the corpora came from large datasets. As shown in Table 2, all models, except ours, use a specific corpus to fine-tune their model to suit the objectives; this makes the process of training BERT-based models utilize high computation resources due to the complexity of the pre-training process and the large data source size. For example, even the original BERT was pre-trained with corpus from the BookCorpus (800M words) [19] and English Wikipedia (2,500M words). Furthermore, the cost of pre-training is relatively expensive; it takes four days on 4 to 16 Cloud TPUs. Even in other contexts, e.g., Twitter-roBERTa [20], which aims to detect Twitter sentiment, uses over 100,000 instances for training and takes 8–9 days on 8 NVIDIA V100 GPUs.

This pre-training task and large data size make the process of training BERT-based models quite expensive. Hence, the techniques we use in this article can be trained faster and use fewer computation resources.

### 1.1.2. Zero-Shot Learning (ZSL)

Generally, zero-shot learning (ZSL) is a task of training a classifier on one set of labels and then evaluating it in a new set of labels that the classifier has never seen before. For example, traditional zero-shot learning requires providing some kind of descriptor for an unseen class [21] (such as a set of visual attributes or simply the class name) in order for a model to be able to predict that class without the training data.

One example for zero-shot learning in NLP is presented by Joe Davison [22], who tested this approach by using Sentence-BERT [23], a novel strategy for producing sequence and label embeddings that fine-tune the pooled BERT sequence representations for greater semantic richness.

Assume we have a sequence embedding model $\Phi_{sent}$ and a set of possible class names $C$ to formalize this. We classify a given sequence $x$ using the following criteria:

$$\hat{c} = \underset{c \in C}{\operatorname{argmax}} \cos(\Phi_{sent}(x), \Phi_{sent}(c)) \qquad (2)$$

where cos is the cosine similarity.

With this approach, Joe Davison can achieve an F1 score of 46.9 on the Yahoo Answers [24] topic classification task. A limitation of this approach is that we need to have some amount of labeled data or annotated data for a subset of the classes that we are focused on.

### 1.1.3. Natural Language Inference (NLI)

In Natural Language Processing (NLP) task, there is one task called Natural Language Inference (NLI). This task is to determine if a "hypothesis" is entailment, contradiction or neutral on a given "premise". By adapting this task to use on zero-shot learning, Yin et al. [25] used the pre-trained Multi-genre NLI (MNLI) sequence-pair classifier as an out-of-the-box zero-shot text classifier, achieving an F1 score of 37.9 on Yahoo Answers using the smallest version of BERT, fine-tuned only on the MNLI corpus. Additionally, Joe Davison was able to reproduce this approach by using a larger model, achieving an F1 score up to 53.7.

Since 1998, around 8000 articles on ScienceDirect have related to zero-shot classification [26]. More specifically, 2233 of these articles were published between 2020 and 2022 [27]. Most of the studies focus on preparing a dataset for zero-shot classification [28] and Computer Vision for classification medical image [28–31]. Despite the enormous number of articles related to this method, the publicly available zero-shot model based on BERT is scarce.

At the time this paper was written, there were another 8,861 models on the huggingface website based on BERT that use different contexts for training [32]. Among nearly 10,000 models, we found only 69 models that could be classified as "zero-shot classification" [33]. So far, there has been only one article from Lupart, S. et al. [34] that used BioBERT with Medical Subject Headings (MeSH) for Zero-shot classification of COVID-19 articles.

Our primary contributions are summarized as follows:

- We tried to improve the SciBERT model by increasing the model vocabulary.
- In this paper, we propose a new method of using the pre-trained BERT model for zero-shot classification of COVID-19 literature that requires only one Tesla V100.
- We demonstrated how our model executes faster and uses fewer computation resources than the comparators.
- Additionally, we demonstrated how all models perform with different GPUs.
- Finally, we demonstrated the performance of all models.

## 2. Materials and Methods

This study was conducted using the diagram shown in Figures 1–3. The data were prepared to train and test our zero-shot classification model to demonstrate how our model is executed against the comparators. Finally, the results were collected and analyzed.

### 2.1. Data Collection

To train the proposed model, we used data from Kaggle website provided by the Allen Institute for AI. As this dataset is updated frequently, in this paper we use data that were published on 19 April 2021 (Version 87) [35]. This dataset has a total of 536,817 articles from PubMed Central (PMC), PubMed, the World Health Organization's COVID-19 Database and the preprint servers bioRxiv, medRxiv, and arXiv [1]. Table 3 shows some examples from the CORD-19 dataset.

**Table 3.** Example data from CORD-19 dataset.

| Column Name | Description | Data |
|---|---|---|
| cord_uid | String valued field that assigns a unique identifier to each CORD-19 paper | ug7v899j |
| sha | String valued field that is the SHA1 of all PDFs associated with the CORD-19 paper | d1aafb70c066a2068b02 786f8929fd9c900897fb |
| source_x | String valued field that is the names of sources | PMC |
| title | String valued field for the paper title | Heterogeneous nuclear ribonucleoprotein A1 regulates RNA synthesis of a cytoplasmic virus |
| doi | String valued field for the paper DOI | 10.1251/bpo66 |
| pmcid | String valued field for the paper's ID on PubMed Central | PMC302190 |
| pubmed_id | Integer valued field for the paper's ID on PubMed | 14702098 |
| license | String valued field with the most permissive license | no-cc |
| abstract | String valued field for the paper's abstract | The UBA domain is a conserved sequence motif among polyubiquitin binding proteins. For the first tim... |
| publish_time | String valued field for the published date of the paper in yyyy-mm-dd format | 2003-12-12 |
| authors | String valued field for the authors of the paper. Each author name is in Last, First Middle format and semicolon-separated | Pridgeon, Julia W.; Geetha, Thangiah; Wooten, Marie W. |
| journal | String valued field for the paper journal | Biol Proced Online |
| mag_id | Integer valued filed for Microsoft Academic Graph, Deprecated | - |
| who_COVIDence_id | String valued field for the ID assigned by the WHO for that paper | #20061721 |
| arxiv_id | String valued field for the arXiv ID of that paper | 2004.09354 |
| pdf_json_files | String valued field containing paths from the root of the current data dump version to the parses of the paper PDFs into JSON format | document_parses/pdf_ json/4eb6e165ee705e2 ae2a24ed2d4e67da428 31ff4a.json |
| pmc_json_files | String valued field corresponding to the full text XML files downloaded from PMC | document_parses/pmc_ json/PMC1481583.xml. json |
| url | String valued field containing all URLs associated with that paper, comma separated | https://www.ncbi.nlm (accessed on 27 April 2022) .nih.gov/pmc/article s/PMC1481583/ |
| s2_id | String valued field containing the Semantic Scholar ID for that paper | 9445722 |

Natural Language Inference (NLI) dataset.

As mentioned earlier, Natural language inference (NLI) takes into account two sentences: a "premise" and a "hypothesis." Given the premise, the aim is to establish if the hypothesis is true (entailment) or false (contradiction). NLI datasets are used to train models using sequence-pair classification when employing transformer topologies such as BERT. Both the premise and the hypothesis are fed into the model as separate segments,

and a classification head predicts one of the terms (contradiction, neutral, or entailment). We use 3 differences NLI datasets for training our model as follows:

### 2.1.1. Stanford Natural Language Inference (SNLI) Corpus

The Stanford Natural Language Inference (SNLI) corpus [36] is a collection of 570k human-written English phrase pairs that have been carefully tagged with the terms entailment, contradiction, and neutral for a balanced classification. Language in the dataset is English, as spoken by the users of the website Flickr and as spoken by crowd workers from Amazon Mechanical Turk. The original datasets are composed of premise, hypothesis, labels from five annotators and the gold label, as shown in Table 4.

**Table 4.** Example data from SNLI corpus.

| Premise | Label | Hypothesis |
|---|---|---|
| A man inspects the uniform of a figure in some East Asian country. | contradiction C C C C C | The man is sleeping. |
| An older and younger man smiling. | neutral N N E N N | Two men are smiling and laughing at the cats playing on the floor. |
| A black race car starts up in front of a crowd of people. | contradiction C C C C C | A man is driving down a lonely road. |

### 2.1.2. Multi-Genre Natural Language Inference (MultiNLI) Corpus

The SNLI dataset has some limitations as the sentences in SNLI are derived from a single text genre—image captions—and are thus limited to descriptions of concrete visual scenes. As a result, the hypothesis sentences used to describe these scenes are short and simple. The MultiNLI [37] corpus contains around 433k hypothesis/premise pairs. It is similar to the SNLI corpus but covers a range of genres of spoken and written text and supports cross-genre evaluation. Examples of this dataset are shown in Table 5.

**Table 5.** Example data from MultiNLI corpus.

| Premise | Label | Hypothesis |
|---|---|---|
| Met my first girlfriend that way. | FACE-TO-FACE contradiction C C N C | I did not meet my first girlfriend until later. |
| 8 million in relief in the form of emergency housing. | GOVERNMENT neutral N N N N | The 8 million dollars for emergency housing was still not enough to solve the problem. |
| Now, as children tend their gardens, they have a new appreciation of their relationship to the land, their cultural heritage, and their community. | LETTERS neutral N N N N | All of the children love working in their gardens. |

### 2.1.3. MedNLI—A Natural Language Inference Dataset for the Clinical Domain

This dataset [38] came from MIMIC-III [16], extracted only the Past Medical History section, and was annotated by four clinicians over six weeks. It contains 14,049 sentence pairs, but the difference between this and the previous two datasets is that the label is only gold, with no individual clinician judgment. Examples from the MedNLI dataset are shown in Table 6.

**Table 6.** Example data from MedNLI corpus.

| Premise | Hypothesis | Label |
|---|---|---|
| ALT, AST, and lactate were elevated as noted above. | Patient has abnormal LFTs. | entailment |
| Chest x-ray showed mild congestive heart failure. | The patient complains of cough. | neutral |
| Aorta is mildly tortuous and calcified. | The aorta is normal. | contradiction |

*2.2. Data Processing*

2.2.1. Vocabulary Preparation

Originally, SciBERT was introduced in 2019 before the outbreak of COVID-19. In order to produce an up-to-date model that covers COVID-19 literature, we decided to add more vocabulary to the model. Unmodified uncased SciBERT has 31,090 vocabularies, which are extracted from a computer science and broad biomedical domain, 18% and 82%, respectively. For our work, we used the uncased version of SciBERT.

The processes of adding new vocabulary to the SciBERT model are as follows:

- The CORD-19 dataset contains raw data from each article with two directories: pdf_json and pmc_json. The pmc_json directory is a directory which has only xml files and does not have an abstract section; as we intended to classify the abstract, we excluded this directory in this process. The pdf_json directory has only json files, which have a metadata section containing an abstract for each file. Eventually we created a pandas dataframe from the json files in the pdf_json directory, then we saved it as a comma-separated values (csv) file. The resulting csv file size was 9.77 GB from 191,569 articles.
- Articles that did not have an abstract and other sections, except an abstract from the file in the previous step, were discarded. The remaining abstracts were converted to lowercase and all special characters were removed (including a newline or \n character). There were 124,979 remaining abstracts.
- Infrequent or too frequent vocabulary, over 10,000 and less than 100 times, were removed. The remaining vocabulary words to add were 12,825.
- Finally, we added the new vocabulary to the SciBERT model, resulting in a total of 33,993 vocabularies in the improved model.

2.2.2. CORD-19 Dataset Preparation

A total of 536,817 articles were extracted from the CORD-19 dataset (Version 87), but there were some records which had neither a title nor abstract. Furthermore, there were articles that did not relate to COVID-19. Therefore, we discarded these records from classification by filtering out the records that did not have both a title and abstract. Because we only selected COVID-19 articles, only articles that were published before 30 November 2019 were included; note that articles that did not contain a publication date also remained in the dataset. The final dataset for classification had a total of 442,657 articles.

2.2.3. Test Dataset Preparation

Since the CORD-19 dataset has no label for each article, in order to measure the model performance, we had to find the label for each article. We found that the LitCovid database [39] had labels. Every article on LitCovid website had one or more labels (General, Mechanism, Transmission, Diagnosis, Treatment, Prevention, Case Report, Forecasting). We wrote a python script to scrape pmid, title, journal and abstract from this website (run from January to February 2021), as illustrated in Figure 1. Table 7 shows the numbers of each label from LitCovid.

**Table 7.** Number of articles in Test dataset based on LitCovid database.

| Label | Number of articles |
|---|---|
| Case Report | 8786 |
| Diagnosis | 10,005 |
| Forecasting | 741 |
| General | 1909 |
| Mechanism | 3405 |
| Prevention | 45,005 |
| Transmission | 3943 |
| Treatment | 36,059 |
| Total | 109,853 * |

* Note no label = 321,174, no title article = 11,630.

**Test data preparation**

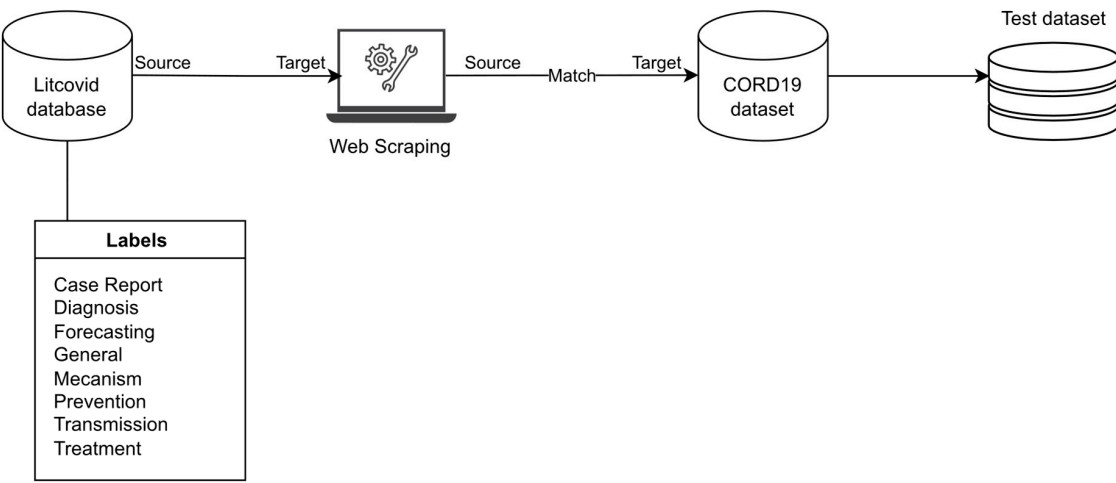

**Figure 1.** Test data preparation process.

2.2.4. NLI Datasets Preparation

- SNLI and MultiNLI Datasets

Both datasets share the same schema. They contain a sentence for premise and hypothesis and a gold label, which is the label chosen by the majority of annotators. We used these three fields to train SciBERT for NLI task after adding new vocabulary. Other fields in these datasets were discarded for the training process as they are not relevant to the NLI task. These include sentence_parse, sentence_binary_parse, annotators_label, ID and genre for MultiNLI dataset. We use only the training part for these two datasets. There are some premises and hypotheses which did not have a gold label, represented with "-" label in the dataset. These type of sentence pairs are 2.0% and 1.8% for SNLI and MultiNLI, respectively, and we excluded them from training. In summary, we acquired training sentence pairs from SNLI and MultiNLI as 549,367 and 392,702 respectively.

- MedNLI Dataset

Unlike the first two NLI Datasets, it contains only the gold label, which came from each clinician, so the labels are only entailment, contradiction or neutral; there was no "-" label in this dataset. The fields in this dataset schema were similar to the previous two datasets. Finally, we acquired 11,232 sentence pairs for training the model.

### 2.3. Training New Downstream Task for SciBERT Model

The original tasks of the SciBERT model [18] are Named Entity Recognition (NER), PICO Extraction (PICO), Text Classification (CLS), Relation Classification (REL) and Dependency Parsing (DEP), which does not include NLI task for building a zero-shot classification model. According to W.Yin et al. [25] and Joe Davison's blog post [22], it is possible to build a model capable of zero-shot classification using the NLI dataset.

We trained SciBERT with a new downstream task (NLI) with the previous three NLI datasets (SNLI, MultiNLI, MedNLI), using only one virtual machine with a spec of 16 virtual CPU cores from Xeon Gold 6244 processor, 32 GB of system memory, 300 GB of virtual storage space, passthrough NVIDIA Tesla V100 and Windows 10 operating system. Other software environments are CUDA toolkits version 11.2, python version 3.8.5 from anaconda3 version 4.9.2, pytorch version 1.9.0, tensorflow version 2.5.0, transformers version 4.8.1. The overall processes are illustrated in Figure 2.

**Training process**

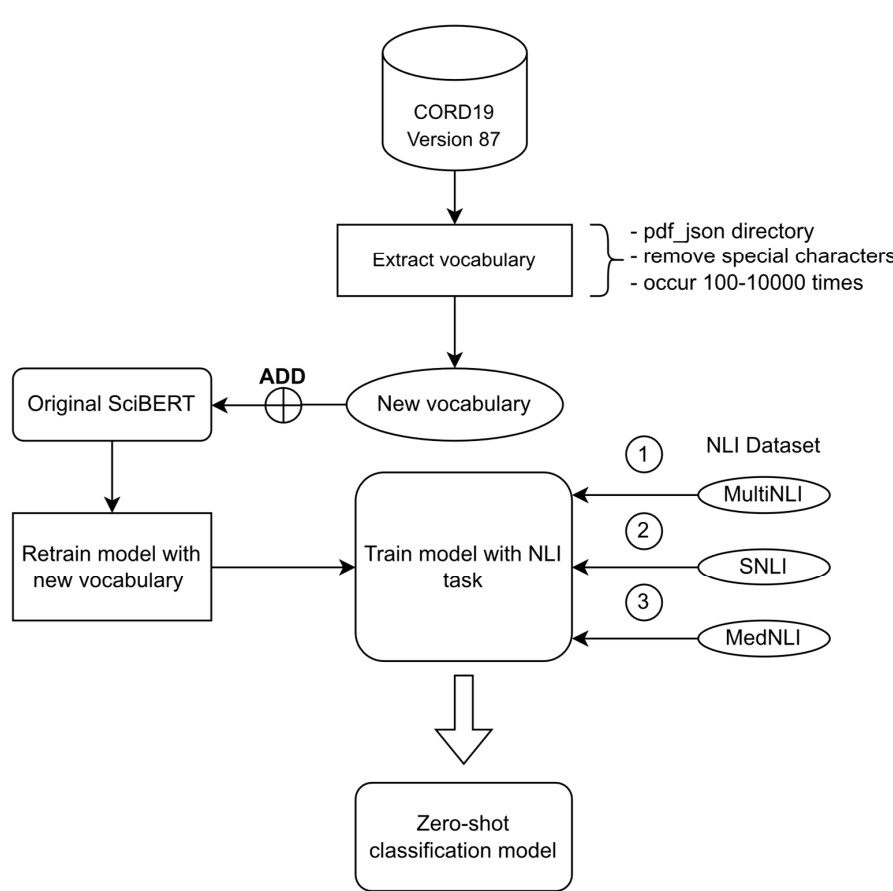

**Figure 2.** Training process for zero-shot classification model from SciBERT.

- Training parameters

To retrain the SciBERT model after adding new vocabulary, we did not change any parameters from the original SciBERT model and only added new vocabulary to the tokenizer. After retraining the model with new vocabulary, we trained SciBERT in the NLI downstream task with SNLI, MultiNLI and MedNLI. The parameters we used to train NLI downstream with SNLI and MutliNLI were similar to each other; the parameters were batch size per device during the training (batch_size) = 16, batch size for evaluation (batch_size_eval) = 64, warmup steps (warmup_steps) = 500, strength of weight decay (weight_decay) = 0.001, learning rate (learning_rate) = $5 \times 10^{-5}$ and 3 epochs training.

However, for MedNLI, we had to adjust these parameters due to a low number of samples (around 10k compared to 400k–500k of sentence pairs). We reduced the batch size for evaluation from 64 down to 4, decreased the learning rate from $5 \times 10^{-5}$ to $1 \times 10^{-6}$ and trained the model for 50 epochs for the MedNLI step.

- Training NLI downstream task process

We utilized the same approach proposed by W. Yin et al. and Joe Davison [22,25]. A summary of all three NLI corpora used to produce SciBERT with zero-shot classification capability is shown in Figure 3.

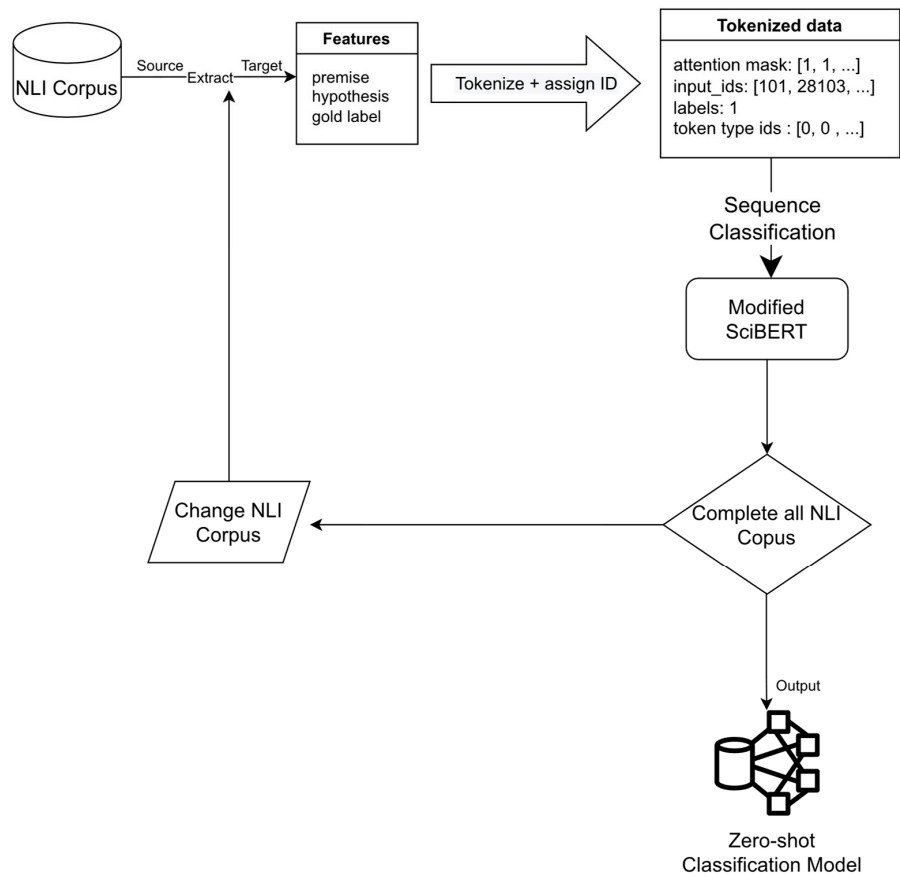

**Figure 3.** The detailed process for making zero-shot classification model.

After adding new vocabulary to the original SciBERT model, we used three NLI corpora to make SciBERT capable of zero-shot classification. First, we extracted features from the NLI corpus, including premise, hypothesis, and gold label. Second, we tokenized the premises and hypotheses into tokens using our model tokenizer. The tokenized result was fed to train the modified SciBERT model with Sequence Classification. Finally, we repeated these processes for all NLI corpora. The final result was the SciBERT model with zero-shot classification ability.

## 3. Results

### 3.1. Model and Metric Comparators

To compare accuracy, time, memory, and space usage, we selected a further three models from the huggingface website: xlm-roberta-large-xnli (xlm-roberta-large) [40], bart-lage-mnli-yahoo-answers (bart-large) [41] from Joe Davison and COVID-Twitter-BERT v2 MNLI (CT-bert V2) [42] from digitalepidemiologylab. The criteria for selecting the

comparators were the model created by Joe Davison that is capable of carrying out "zero-shot classification" on the huggingface website (xlm-roberat-large, bart-large) or model training in a COVID-19 context (CT-bert V2).

All models were used for zero-shot classification via a pipeline in the transformers package. The results were saved in text files for calculating accuracy, precision, recall, and f1-score.

The accuracy of each model was calculated by comparing the true label from LitCovid database and the predicted label from the transformers pipeline using parameter multi-class = True. When this parameter is set to True, the labels are classified as independent and the probabilities for each candidate are normalized using a softmax of the entailment score vs. the contradiction score.

For time and memory usage comparison, when executing each model on CORD-19 dataset, we executed four tests based on different GPUs as following TitanXp (12 GB of video memory), TitanV (12 GB of video memory), Tesla P100 (12 GB of video memory), Tesla V100 (with 8 GB video memory from the virtual machine) and Tesla V100 (with 32 GB video memory from passthrough virtual machine). Figure 4 illustrates how the experiment was set up and how the output was determined.

**Experiment setup**

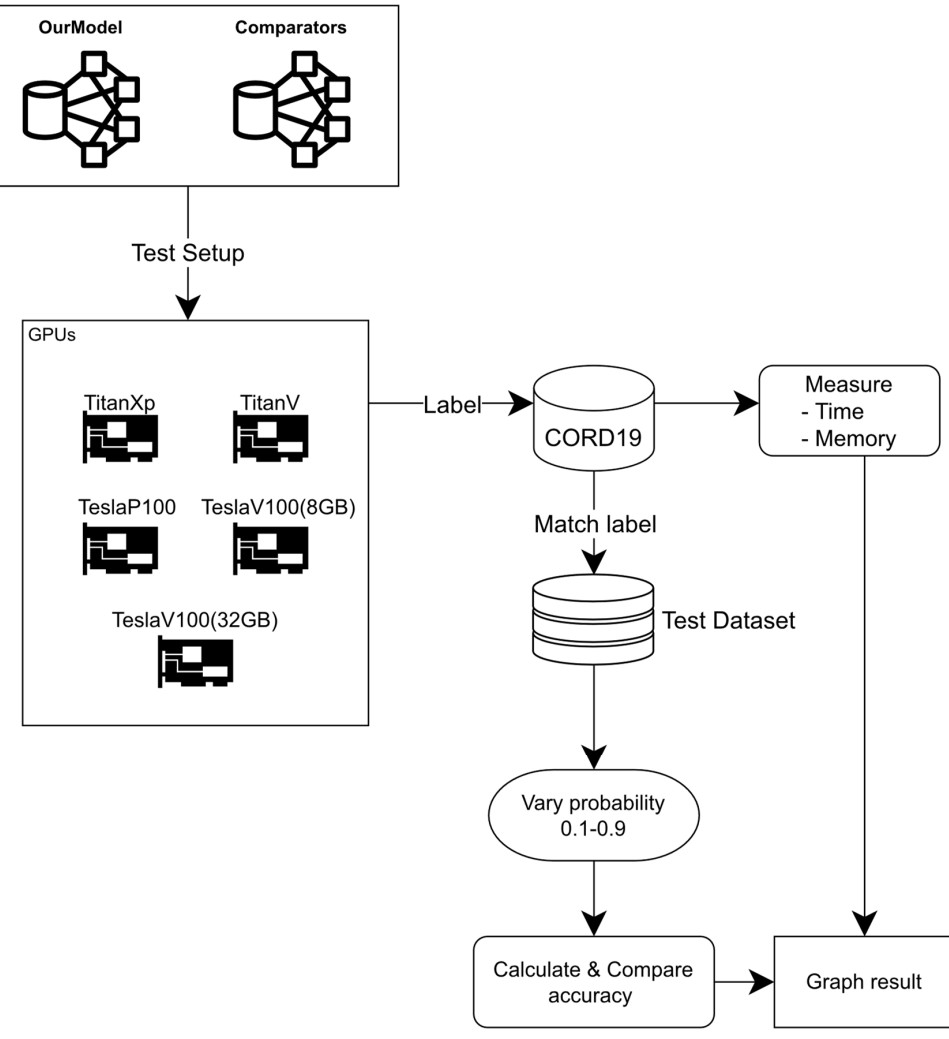

**Figure 4.** Experiment setup for our study.

### 3.2. Model Training Time

The time used in each step of training is presented in Table 8.

**Table 8.** Time usage in each step of the training zero-shot classification model.

| Step | Time Usage |
|---|---|
| Retrain SciBERT after adding new vocabulary | 52 min 26 s |
| Train NLI downstream task with dataset | |
| • MultiNLI | 2 h 26 min 41 s |
| • SNLI | 2 h 56 min |
| • MedNLI | 1 h 15 min 20 s |
| Total training time | 7 h 30 min 27 s |

As shown in Table 8, the training time in all steps was approximately 1 to 3 h, depending on the dataset.

### 3.3. Model Size

The models' size after training completion is shown in Table 9, also including the model vocabulary number that was embedded in each model. The biggest model in this experiment is xlm-roberta-large, followed by bart-large, our model and CT-bert V2. It should be noted that when comparing model size and vocabulary number, our model is comparable to CT-bert V2 in both model size and vocabulary.

**Table 9.** Comparison of model and vocabulary size.

| Model | Size (GB) | Size (Byte) | Vocabulary |
|---|---|---|---|
| SciBERT uncased | 0.41 | 440,457,270 | 31,090 |
| Our Model | 1.25 | 1,348,305,925 | 33,993 |
| CT-bert V2 | 1.24 | 1,341,435,039 | 30,522 |
| xlm-roberta-large | 2.09 | 2,253,900,123 | 250,002 |
| bart-large | 1.51 | 1,630,968,591 | 50,265 |

### 3.4. Token Volume

From a total of 442,657 articles in the CORD-19 database, we tokenized all abstracts with a tokenizer from each model. The total ID and number of tokens are presented in Table 10.

**Table 10.** Token volume from all models.

| Model | Number of Token ID | Total Number of Tokens |
|---|---|---|
| SciBERT uncased | 26,168 | 57,183,423 |
| Ourmodel | 26,783 | 61,326,123 |
| CT-bert V2 | 26,663 | 61,209,919 |
| xlm-roberta-large | 60,535 | 65,282,093 |
| bart-large | 46,242 | 59,891,876 |

From Table 10, we can observe that our model and CT-bert V2 give the same amount of both number of ID and tokens. This is the same phenomenon as seen earlier in the model size comparison. Additionally noted is that even bart-large has twice the token ID numbers compared to our model and CT-bert V2, but the total number of tokens is lesser.

### 3.5. Time Usage

The time taken to load each model to GPU memory and the total execution time for each model from loading until all 442,657 abstract articles had finished being classified were measured. The results are shown in Table 11 and Figures 5 and 6.

**Table 11.** Loading time and total execution time for all models on different GPUs.

| | Loading Time (Seconds) | | | | |
|---|---|---|---|---|---|
| Model | GPUs | | | | |
| | TitanXp | TitanV | TeslaP100 | TeslaV100 (8 GB) | TeslaV100 (32 GB) |
| Ourmodel | 87.7915 | 76.41922 | 64.9150097 | 75.2837092 | 66.44118 |
| CT-bert V2 | 31.89718 | 14.35474 | 15.7923569 | 18.1387475 | 16.12749 |
| xlm-roberta-large | 61.74109 | 26.52093 | 22.6739579 | 23.0446838 | 22.43863 |
| bart-large | 61.16165 | 18.49952 | 24.0774501 | 21.7254039 | 17.9323 |
| | Total Execute Time (Seconds) | | | | |
| Model | GPUs | | | | |
| | TitanXp | TitanV | TeslaP100 | TeslaV100 (8 GB) | TeslaV100 (32 GB) |
| Ourmodel | 53,513.01 | 52,305.9 | 33,367.07 | 29,430.7 | 28,462.6 |
| CT-bert V2 | 52,498.73 | 41,502.69 | 60,490.08 | 39,057.64 | 36,114.47 |
| xlm-roberta-large | 56,891.2 | 44,877.2 | 66,451.43 | 41,859.77 | 39,318.24 |
| bart-large | 65,056.13 | 50,333.32 | 74,179.42 | 123,832.6 | 43,756.65 |

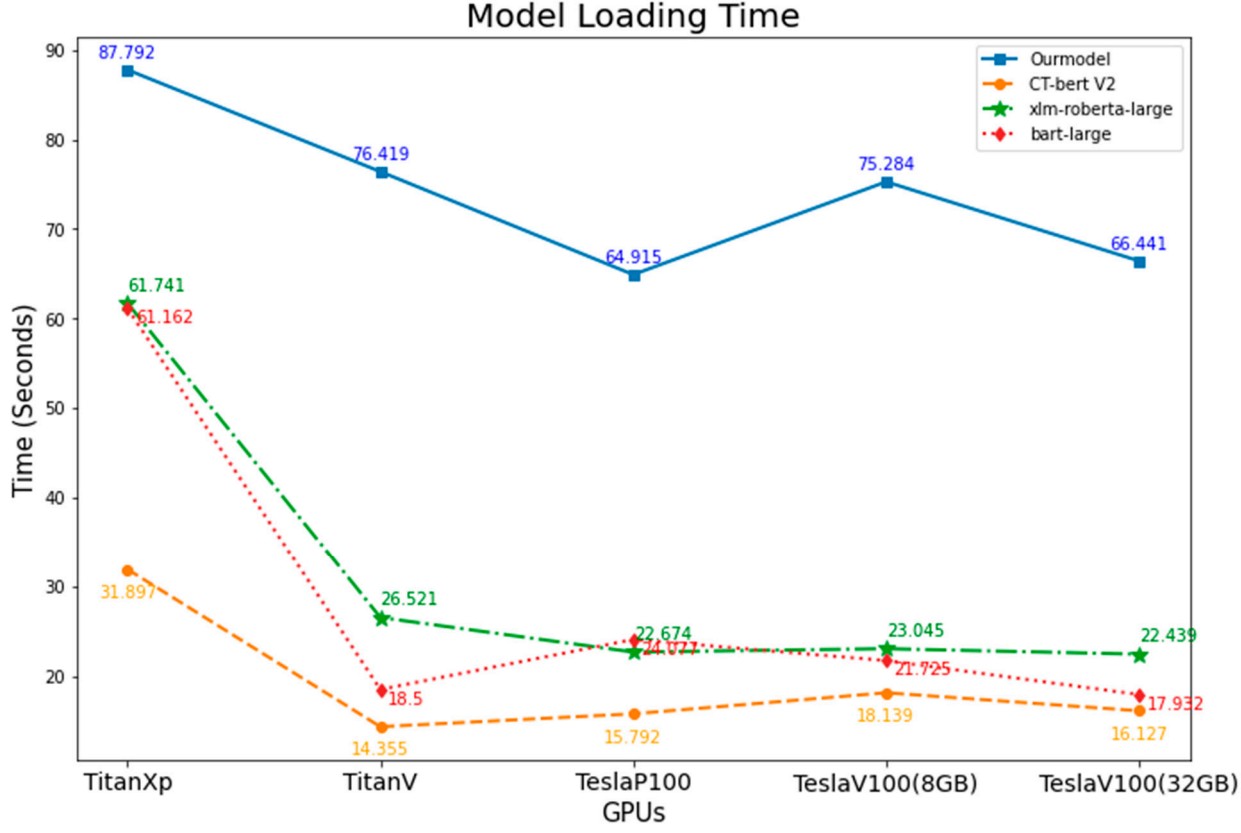

**Figure 5.** Comparison of models' loading time on different GPUs.

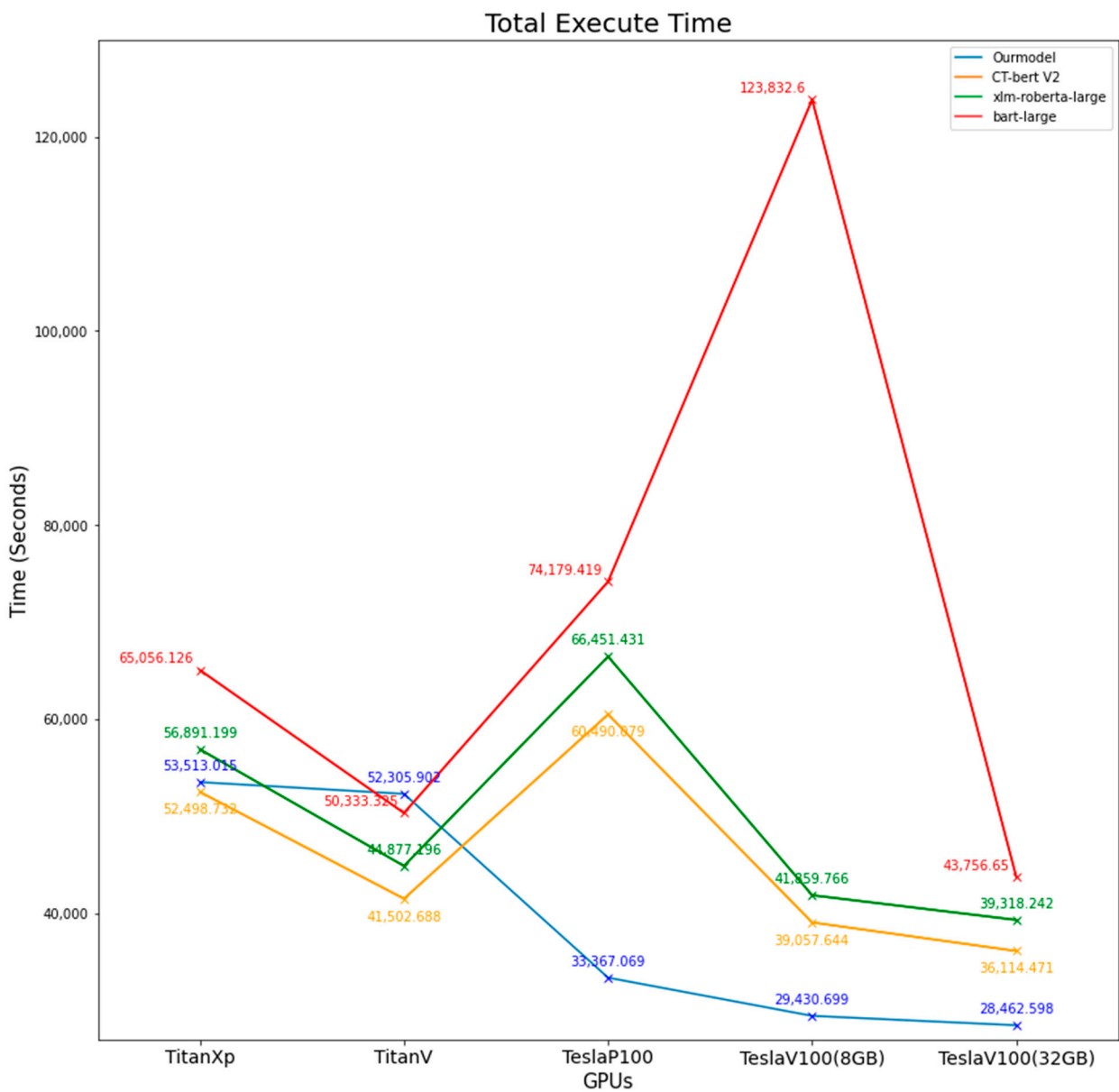

**Figure 6.** Comparison of total execution time from all models on different GPUs.

When we consider model loading time, our model took the longest time compared to other models. However, when considering total execution time, our model took less time on Tesla GPUs; other models had the same result unless executed on TeslaV100 (8 GB). On Titan GPUs, the results were mixed, but there is a trend that when using a higher model (TitanV vs. TitanXp), the model will take less loading and total execution time. On TeslaP100 GPUs, all models had faster loading time when compared with TitanXp, which had the same architecture, but when considering total execution time, TeslaP100 took a longer time than TitanXp. This can be explained by the fact that TeslaP100 uses High Bandwidth Memory (HBM2) instead of GDDR5X on TitanXp, which has more bandwidth [43,44]. On the other hand, TeslaP100 took a longer total execution time than TitanXp on all models except our model. This could be explained by the fact that TeslaP100 has a lower CUDA core count and has lower GPU core frequency [45]. For the outlier, execute bart-large on TeslaV100 (8 GB), the system resource may be utilized by other users who share the same GPU on a virtual machine. We tried to execute bart-large on this system multiple times, but it failed to complete; only the first time were we able to measure the total execution time.

### 3.6. GPUs Memory Usage

In terms of GPUs memory, all models demonstrate identical results, as shown in Figure 7. Our model, on all GPUs, had the least memory usage, followed by CT-bert V2, xlm-roberta-large and bart-large. To summarize, our model approximately used 1.7 to 2.9 GB, CT-bert V2 used around 2.6 to 4 GB, xlm-roberta-large used around 3.8 to 5 GB, and bart-large used around 6.8 to 7.5 GB. Considering bart-large on TeslaV100 (8 GB), the model used nearly 8GB; this could explain why this model failed to complete the task multiple times and had the longest total execution time.

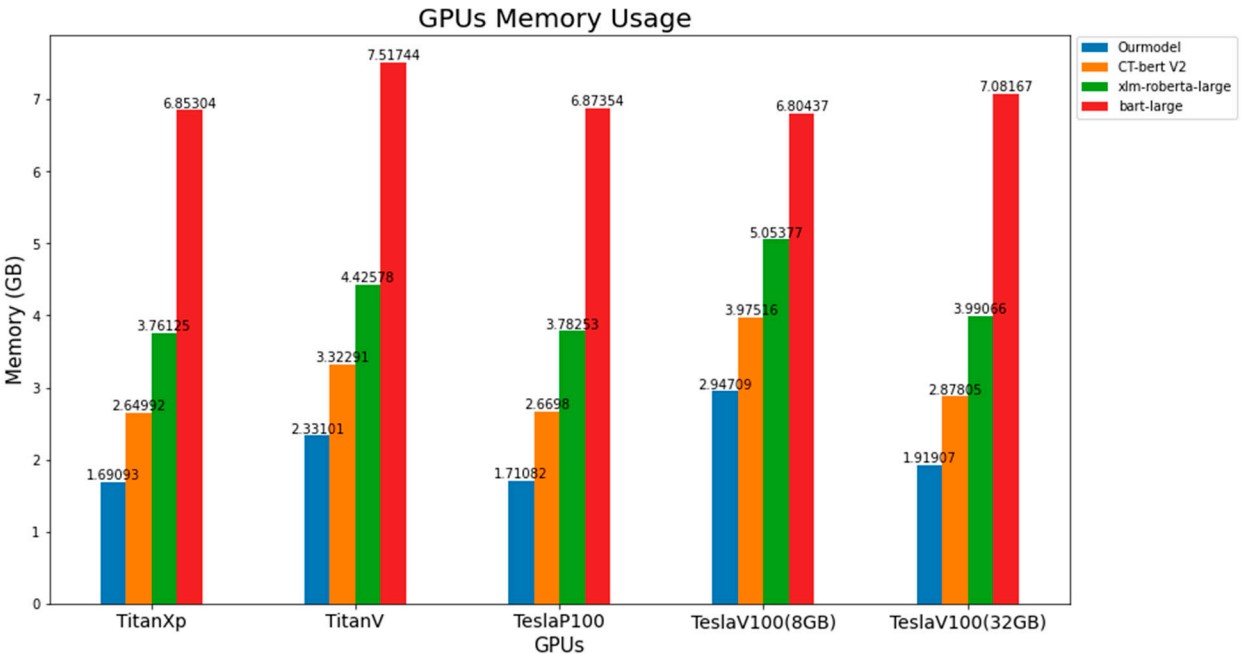

**Figure 7.** Comparison of GPU memory usage of all models on different GPUs.

### 3.7. Precision, Recall and F1-Score

Table 12 shows the metrics from all the models. For precision, the model that had the highest precision on the most of all classes was bart-large. Xlm-roberta-large had the highest score for both recall and f1-score in the most of all classes. CT-bert V2 and our model shared some of the highest scores on these metrices. Our model had comparable precision to other models on diagnosis, forecasting, and treatment classes. For recall metric, our model had a comparable score on case report, diagnosis, mechanism, prevention, and transmission classes. Lastly, for the f1-score, our model achieved comparable scores on case report, diagnosis, mechanism, and prevention classes.

**Table 12.** Precision, Recall and F1-score for all models. (Bolds indicates highest value on that class).

| Class | Precision | | | | Recall | | | | f1-Score | | | | Support |
|---|---|---|---|---|---|---|---|---|---|---|---|---|---|
| | Roberta-large * | Bart-large | CT-Bert V2 | OurModel | Roberta-large * | Bart-large | CT-Bert V2 | OurModel | Roberta-Large * | Bart-large | CT-Bert V2 | OurModel | |
| Case Report | 0.5056 | **0.5823** | 0.3347 | 0.2537 | **0.4919** | 0.1007 | 0.4319 | 0.4646 | **0.4987** | 0.1717 | 0.3772 | 0.3282 | 8,780 |
| Diagnosis | 0.3134 | 0.2139 | **0.3361** | 0.3006 | 0.4186 | **0.8471** | 0.273 | 0.3766 | **0.3585** | 0.3415 | 0.3013 | 0.3343 | 10,005 |
| Forecasting | 0.0442 | **0.2351** | 0.0884 | 0.2065 | **0.5965** | 0.1174 | 0.3954 | 0.0432 | 0.0823 | **0.1566** | 0.1445 | 0.0714 | 741 |
| General | 0.0471 | **0.117** | 0.0657 | 0.0116 | **0.098** | 0.0314 | 0.0115 | 0.0073 | **0.0636** | 0.0495 | 0.0196 | 0.009 | 1909 |
| Mechanism | 0.0558 | **0.0852** | 0.0453 | 0.0361 | 0.4292 | 0.0783 | **0.6799** | 0.4139 | **0.0988** | 0.0816 | 0.0849 | 0.0664 | 3,399 |
| Prevention | 0.6163 | **0.7379** | 0.7374 | 0.4443 | **0.314** | 0.1012 | 0.0978 | 0.1673 | **0.416** | 0.178 | 0.1727 | 0.2431 | 44,958 |
| Transmission | 0.1631 | 0.1247 | **0.2393** | 0.0879 | 0.4415 | **0.8272** | 0.455 | 0.4958 | 0.2382 | 0.2167 | **0.3137** | 0.1493 | 3,941 |
| Treatment | 0.4985 | **0.5292** | 0.4677 | 0.4403 | 0.196 | **0.4743** | 0.2879 | 0.0199 | 0.2813 | **0.5002** | 0.3564 | 0.038 | 36,029 |

* Abbreviation: Roberta-large = xlm-roberta-large.

### 3.8. Accuracy

We varied the predicted class probability for every model from 0.1 to 0.95, except our model, in which we changed the probability from 0.95 to 0.99. Figures 8 and 9 show the accuracy based on probability for all models. All models exhibited the same pattern as increasing probability gave more accuracy, but only for bart-large model did the accuracy rise slower compared to other models. We have drawn a vertical dash red line for each graph to illustrate the elbow point for all models. The elbow point from all models is at probability of 0.80, except for our model, which is 0.90. Considering the probability of 0.80, the xlm-roberta-large model has the highest accuracy of 0.34573, followed by 0.33232 for bart-large, 0.30889 for CT-bert V2, and 0.24551 for our model. For the probability of 0.95, the highest accuracy was seen in the bart-large model, which is 0.49605, followed by 0.4676 for CT-bert V2, 0.39688 for xlm-roberta-large, and 0.30853 for our model (not shown in Figure 8d). Our model has an accuracy of 0.278355 at the elbow point, and the highest accuracy is 0.38388 at a probability of 0.99.

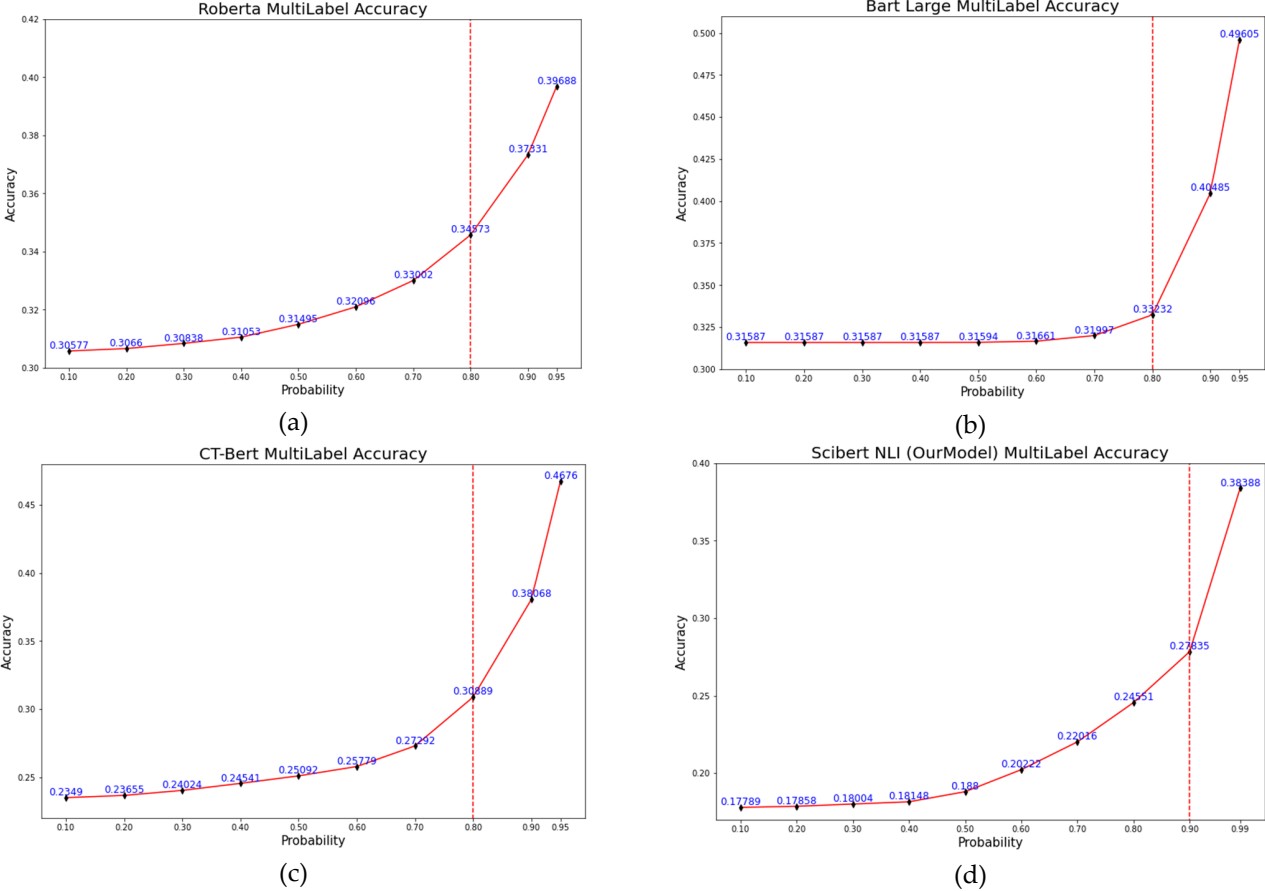

**Figure 8.** Accuracy vs. Probability graph for all models: (**a**) xlm-robera-large, (**b**) bart-large, (**c**) CT-bert V2, (**d**) Our Model.

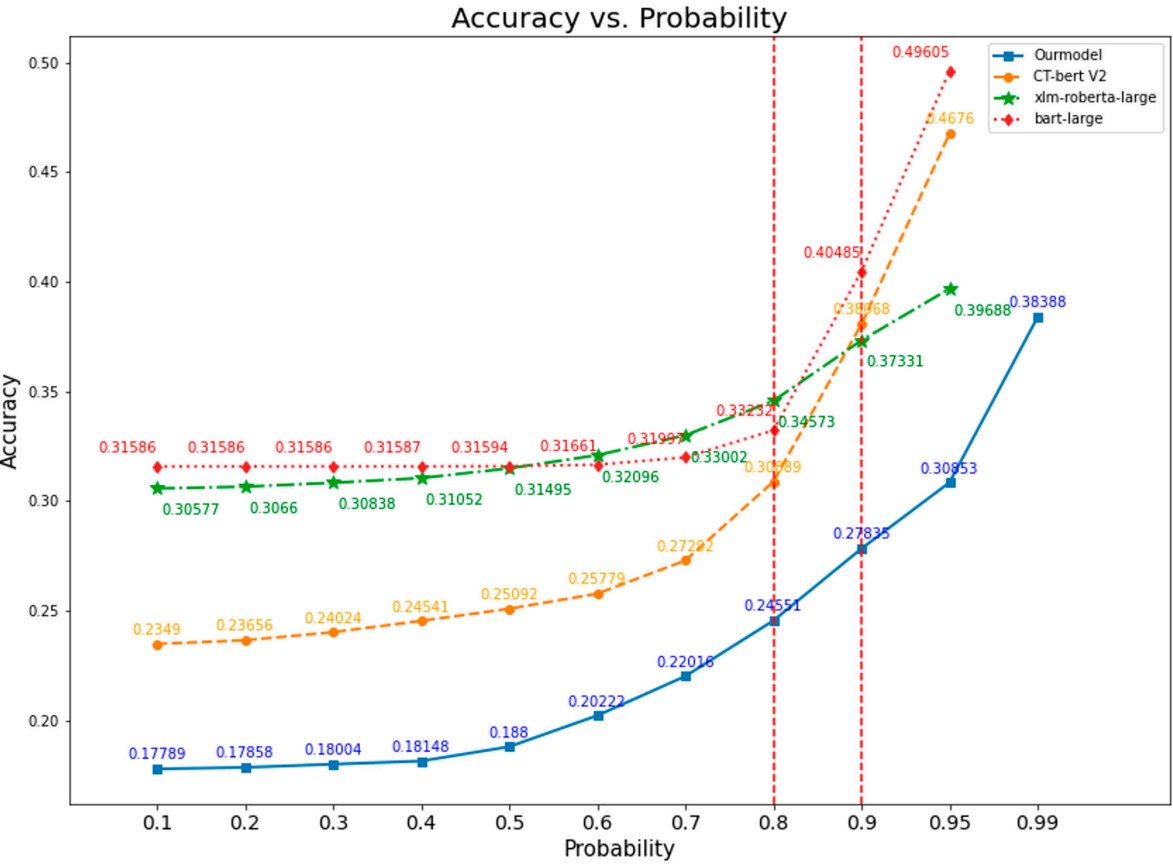

**Figure 9.** Accuracy vs. Probability comparison graph for all models.

## 4. Discussion

In this article, we have successfully built a zero-shot classification model from the original SciBERT by training the model with NLI downstream task from three NLI datasets; MultiNLI, SNLI and MedNLI. Typically, when a machine learning model with a specific task is needed, we have to train a model from the ground up. Examples are Scibert, BioBERT, PubMedBERT, and Clinical BERT. SciBERT, which uses SCIVOCAB, was trained from scratch and used a single TPU v3 with eight cores for one week [18]. BioBERT, which was specifically trained on PubMed abstracts and PubMed Central full articles, was trained on 8 Nvidia V100 GPUs for 23 days [17]. PubMedBERT, which also uses corpus from PubMed, was trained on one DGX-2 machine (16 Nvidia V100 GPUs) for five days [13]. Clinical BERT, which is based on BioBERT and uses corpus from the MIMIC database [16] for downstream task, took around 17–18 days of training on a single Nvidia Titan X GPU. For our comparators in this article, RoBERTa used 1024 V100 GPUs for approximately one day [11]; BART did not specify the number and type of GPUs used for training in the original article [46]. However, we found information on GitHub that training BART took around 11–12 days on 256 GPUs [47]; COVID-Twitter-BERT specifies only that the researcher used Google Cloud TPUs and the training time was on par with BERT-LARGE [48] and BERT-LARGE can be trained on 16 TPUs V3 for 81.4 h [49]. Our model was trained on one Nvidia V100 GPU and took around 7.5 h. If we compare it to a similar model, Clinical BERT, our model performs 57.6 times faster than its counterpart.

Regarding storage space, our model uses more storage space, about three times than the original SciBERT. However, our model uses a comparable or less amount of space when compared with our comparators. This is probably one reason our model works as fast as possible, when considering total execute time. Model loading time is one of the components of total execution time; even though our model took the longest time in this step, it had little effect on the total execution time. The longer loading time could be explained by

the fact that we used the SciBERT model, which does not have NLI for downstream task, and we added this task to the model. When loading the model into GPU memory, the transformers package has to load new NLI downstream task after finishing loading original downstream tasks.

For GPUs memory usage, larger models use more memory than smaller models. The exception is our model, whose model size is close to CT-bert V2 but uses the least GPU memory. We also noticed that when executing zero-shot classification pipeline on other models, the GPUs utilization spike was nearly 100%, but GPU utilization from our model was around 40–50%. One possible explanation is that our model did not build from scratch with the NLI downstream task; when executing the pipeline for zero-shot classification, transformers load only that part to perform.

Accuracy of all models is below 50%, which is around 27–49%. Confusion matrices are illustrated in Figure 10. Every model has at least one class that has been misclassified. A possible explanation for this phenomenon is each abstract in the CORD-19 dataset may have more than one label on the LitCovid database. However, at the time we ran web scraping, we only collected one of them. Classes that co-occur the most on LitCovid database are diagnosis, mechanism, and treatment. These three classes are also the majority in the CORD-19 dataset.

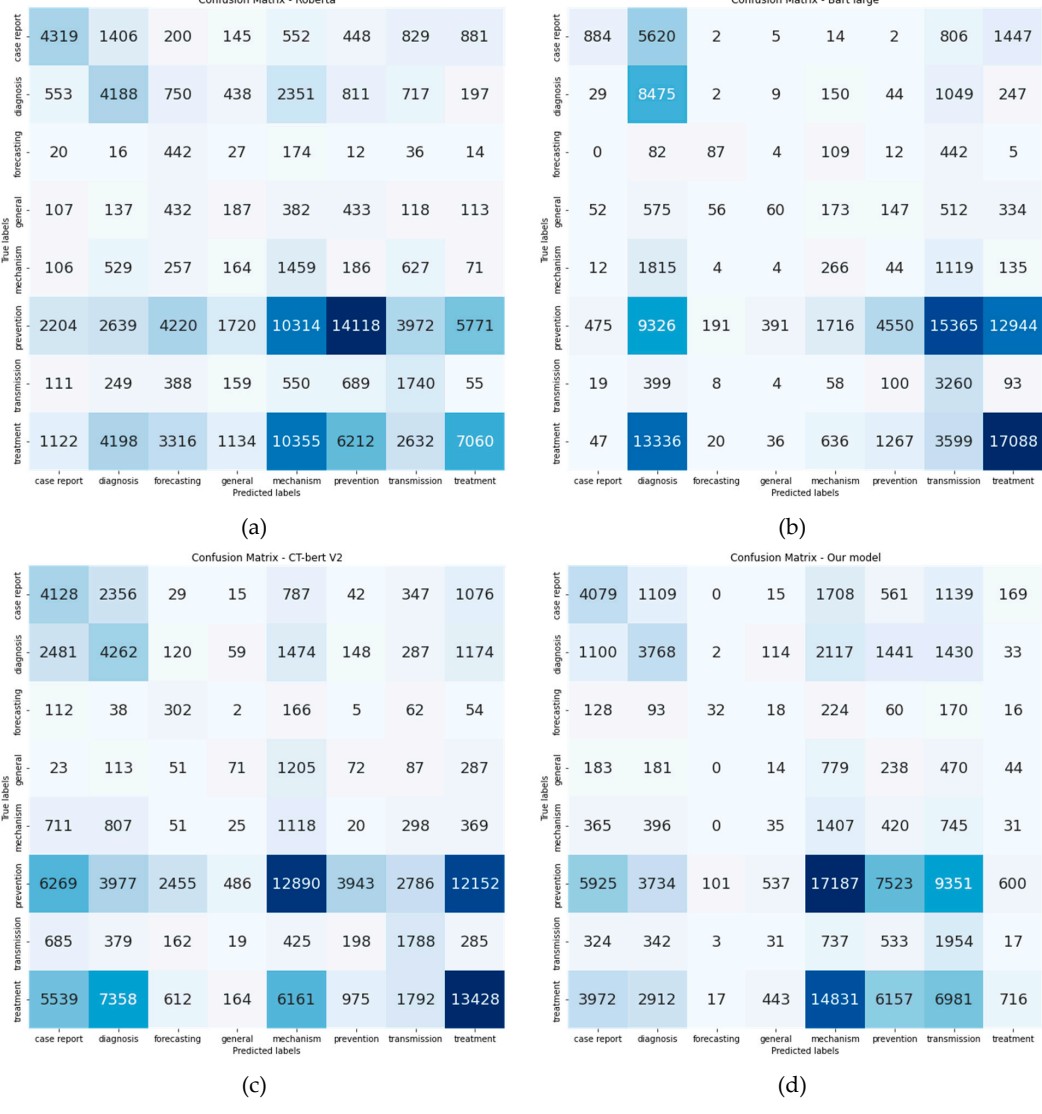

**Figure 10.** Confusion matrix from all models: (**a**) xlm-roberta-large, (**b**) bart-large, (**c**) CT-bert V2, (**d**) Our model.

The two models that have the highest precision, recall, f1-score and accuracy, are bart-large and xlm-roberta-large, respectively. This may be due to vocabulary size and tokenizer. First, a model that has a larger vocabulary size gives higher accuracy. Second, Figure 10 illustrates an overview of each model token. CT-bert V2 and our model token contain sub-words (example are ##vid, ##mic, ##s on Figure 11a,b), which can be explained by the fact that both use the WordPiece tokenizer [50] which breaks new unknown words into sub-words, while bart-large and xlm-roberta-large consider those words together as a sentence. Figure 11c,d show that there is no sub word in the tokens. Table 13 shows the tokenized output from all models. We can observe that each model has different tokens, even from the same word. Another explanation as to why bart-large has the best accuracy, even with the smallest number of tokens, is the model's architecture. While the other three models use BERT architecture, bart-large uses both the Bidirectional encoder and the AutoRegressive decoder [46].

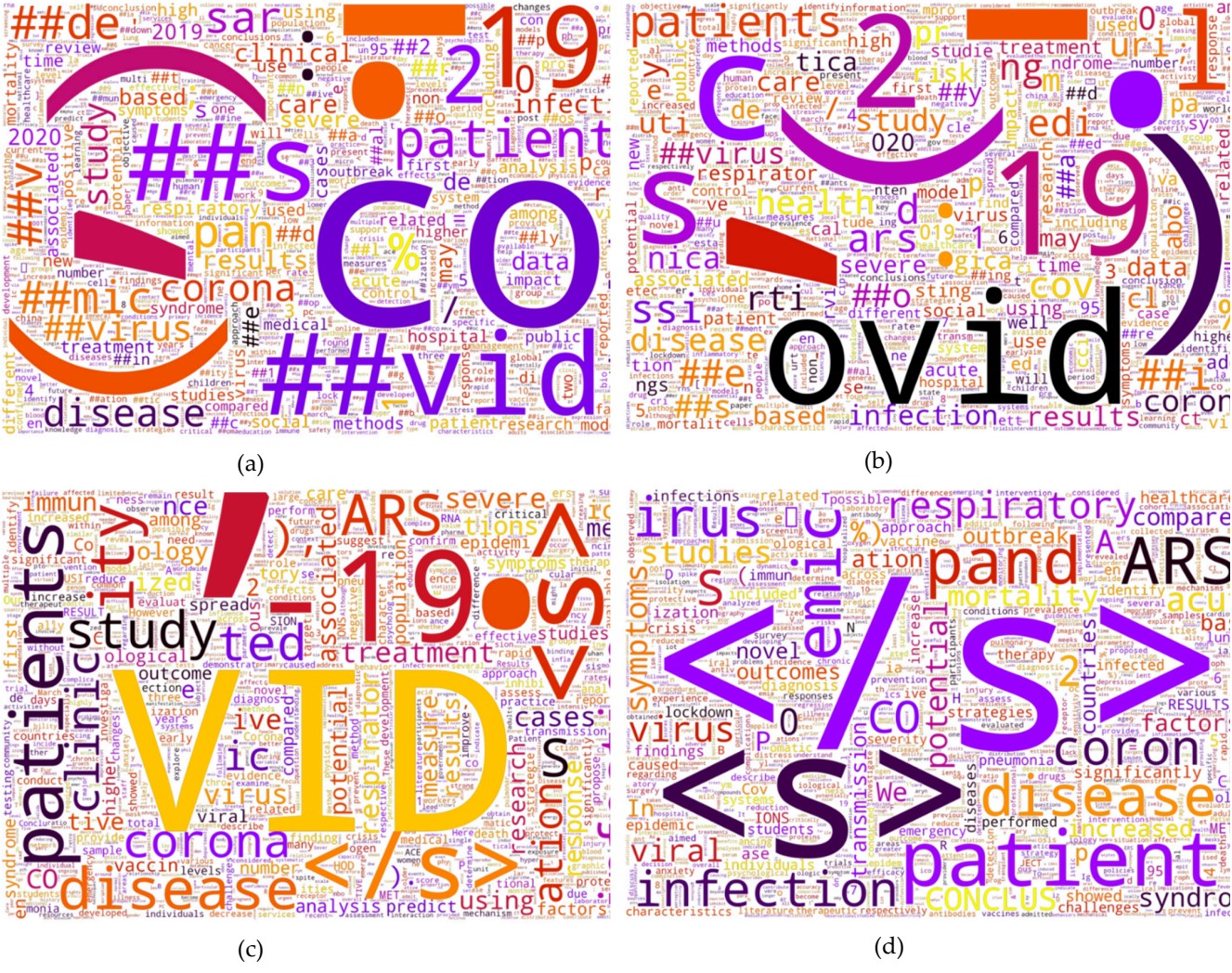

**Figure 11.** Word cloud from each model token after removing stop word and special tokens: (**a**) CT-bert V2 token, (**b**) Our model token, (**c**) xlm-robera-large token, (**d**) bart-large token.

**Table 13.** Example of Tokenized output from all models.

| Word | CT-Bert V2 | Our Model | Xlm-Roberta-Large | Bart-Large |
|------|-----------|-----------|-------------------|------------|
| **COVID-19** | co, ##vid, -, 19 | c, ovid, -, 19 | co, vid, -19 | c, ov, id, -, 19 |
| **respiratory** | respiratory | respirator, y | respirator, y | res, pir, atory |
| **coronavirus** | corona, ##virus | corona, ##virus | corona, virus | cor, on, av, irus |
| **pandemic** | pan, ##de, ##mic | pandemic | pande, mic | p, and, emic |
| **wuhan** | wu, ##han | wuhan | w, uhan | w, uh, an |
| **virus** | virus | virus | virus | v, irus |
| **lopinavir** | lo, ##pina, ##vir | lopinavir | lo, pina, vir | l, op, inav, ir |

There are a few previous studies that have the same objective as our study, classified in the CORD-19 dataset. A study from 2020 [51] used the LitCovid dataset to train the machine learning model. In this article, they trained Logistic Regression and Support Vector Machine for the traditional machine learning model, LSTM model for the neural network model and fine-tuned the BioBERT, Longformer [52] for the BERT based model. Accuracy and f1-score from all models on the LitCovid test set are quite impressive; the accuracy ranged from 68.5 to 75.2% and f1-score ranged from 81.4 to 86.2%. However, when they changed the test set to the labelled CORD-19 test set, the accuracy and f1-score dropped dramatically, accuracy dropped to 29.0 to 41.3% and the f1-score dropped to 62.9 to 70.0%, which is similar to our study. They also found that forecasting and prevention classes were more overlapped than in other classes.

The study from 2021 [53] evaluated fifteen transformer-based models to detect misinformation in several COVID-19 datasets, which are COVID-CQ [54], CoAID [55], ReCOVery [56], CMU-MisCov19 [57], and COVID19FN [58]. In this study, tokenizers and models tailored to COVID-19 data did not outperform general-purpose tokenizers and models, according to the researchers. Our study result is similar to this study, as our model and CT-bert V2 did not overcome xlm-roberat-large and bart-large.

Several unsupervised machine learning techniques require lesser computation resources, such as clustering. Clustering requires lesser resources and can be carried out by using only the CPU. Nevertheless, whether the result is correct or not cannot be determined, as the label is generated based on similarity. Our approach is based on BERT architecture, a deep learning technique, thus the accuracy of the classification result can be calculated. Additionally, clustering on the CORD-19 dataset has been carried out in various other studies since 2020 [59], so it would not be reasonable to replicate these investigations.

*Limitations of Our Study*

This paper has some limitations, including:

1. We used CORD-19 dataset version 87 to add new vocabulary to the SciBERT model. The CORD-19 dataset is updated regularly; now, it is version 105. Therefore, the dataset will now contain more articles than the version used in this paper.
2. Our model can understand English only; the dataset may contain articles from different languages, meaning that our model cannot correctly classify them. On the other hand, xlm-roberta-large-xnli can understand languages other than English, including: French, Spanish, German, Greek, Bulgarian, Russian, Turkish, Arabic, Vietnamese, Thai, Chinese, Hindi, Swahili, Urdu, etc. [60].
3. The metric that we presented was calculated with one label. However, as mentioned above, some articles were labeled with more than one class, and we did not have the tools to calculate the accuracy of all the labels.
4. Due to limited resources, we could only completely execute the experiment on a virtual machine with Tesla V100 (8 GB) once. After that, the experiment on this virtual machine was interrupted before finishing.

5. The tokenized abstract, which is longer than 512, was trimmed due to the original BERT's length restriction.

## 5. Conclusions and Future Work

We have created a zero-shot classification model based on SciBERT. Our method in this study did not involve building a new model from scratch but involved re-training a SciBERT model with new vocabulary from the CORD-19 dataset and developing a new downstream task from three NLI datasets. This method can save a huge amount of time for training BERT-based models and the final model had comparable results with the comparators. We can identify that using a tokenizer that is more specific to COVID-19 did not improve the accuracy and other metrics. SciBERT was built from both biomedical and computer science corpora and the vocabulary in the SciBERT overlapped the BERT vocabulary by 42%. This indicated that SciBERT vocabulary is more specific to science that to general areas of study. Future studies may involve changing the SciBERT model to a more general-purpose model before adding NLI downstream task for zero-shot classification and consider adding more frequent words (over 10,000 times) into the model vocabulary for more generalized purposes.

**Author Contributions:** Conceptualization, N.N. and P.S.; Data curation, N.N.; Formal analysis, N.N.; Investigation, N.N.; Methodology, N.N. and P.S.; Project administration, N.N.; Resources, N.N.; Software, N.N.; Supervision, P.S.; Validation, N.N.; Visualization, N.N.; Writing—original draft, N.N.; Writing—review and editing, P.S. All authors have read and agreed to the published version of the manuscript.

**Funding:** This research received no external funding.

**Data Availability Statement:** The dataset utilized in this study is available at https://www.kaggle.com/datasets/allen-institute-for-ai/CORD-19-research-challenge (version 87) accessed on 18 April 2020, All model comparators available at https://www.huggingface.co accessed on 26 March 2022, SNLI dataset available at https://nlp.stanford.edu/projects/snli/ accessed on 12 October 2021, MultiNLI dataset available at https://cims.nyu.edu/~sbowman/multinli/ accessed on 12 October 2021, MedNLI dataset available at https://physionet.org/content/mednli/1.0.0/ (need credential access) accessed on 12 October 2021.

**Acknowledgments:** The authors wish to thank Associate Professor Juggapong Natwichai for his guidance and all supports.

**Conflicts of Interest:** The authors declare no conflict of interest.

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
