# Peer review of "Do We Need a Specific Corpus and Multiple High-Performance GPUs for Training the BERT Model? An Experiment on COVID-19 Dataset"

_make, doi:10.3390/make4030030_

Round 1

Reviewer 1 Report

This manuscript proposes to use the zero-shot learning method based on pre-trained BERT to perform classification task for COVID-19 literature. The authors aim to prove that, using zero-shot learning and fine-tuning on existing data and labels (NLI task in this manuscript) can be sufficient to perform other downstream task (publication classification task), and it can save time and computing resources. The manuscript presents many detailed evaluation details which can be practical and helpful for readers. However, the major concern is that the different parts of the manuscripts are disconnected without a clearly defined research gap and good presentation, such as:

    • The authors review a lot of related work in the background, including attention mechanism, BERT and its variants and zero-shot learning. However, there is not enough synthesis of those work to clearly define a gap and research question. Is the gap "zero-shot learning to save training time and computation for biomedical NLP understudied" or other? More synthesis of related work need to be added.
    • Zero-shot learning task needs to be further described and defined in the method section.
    • There are many tables and figures, which are helpful but can be improved. For example, it would be a more direct comparison if four diagrams are plotted in one coordinate system。
    • Is 27.84% accuracy good enough to prove zero-shot learning's success (or considered as comparable)?Would it be possible for some more light-weighted or unsupervised algorithms, such as clustering, outperform this model?
    • The writing need to be largely improved. Some descriptions are confusing (e.g., page #7 line #169: "33993 vocabulary")

Author Response

We have answered the question and comment from yours in the attached pdf file.

Reviewer 2 Report

Paper is well written and described. However, the research is to use BERT on Codie-19 dataset. But they contributed  by developing new vocabulary dataset to improve accuracy of model.

Author Response

(The authors gave the same response as above.)

Reviewer 3 Report

The paper presents the method to make an unsupervised model called the zero-shot classification model from a pre-trained BERT model. The paper is well written but does not contain any related work related to BERT model. Add a related work section with a clear comparison of your contribution and published works.

Results setup needs to be explained before section 3.1. How each step of the training model is being proposed?

Also, what were the selection criteria for the model size?

Limitations must be added in the discussion section

Author Response

(The authors gave the same response as above.)

Round 2

Reviewer 1 Report

The major concerns have been largely resolved. The Introduction section can be further improved, particularly the tables that summarize BERT variants (Table 1, 2 and 3). First the descriptions can be clearer (e.g., Table 2 "Full term", "Summary"),  Additionally, those 3 tables along with their descriptions in manuscript have a significant amount of details, but a large part of those details are not closely associated with the research question, i.e., the how the proposed model is different from them and saves the training time and computation. It would be better to trim and synthesize the related work into a shorter version that's the most precisely associated with research question.

Author Response

Please see the point by point response is in the attached PDF file.
